# Increasing Disaster Medical Assistance Teams’ Intent to Engage with Specific Hazards

**DOI:** 10.3390/ijerph182111630

**Published:** 2021-11-05

**Authors:** Keita Iyama, Takeyasu Kakamu, Kazunori Yamashita, Yoshinobu Sato, Jiro Shimada, Osamu Tasaki, Arifumi Hasegawa

**Affiliations:** 1Department of Radiation Disaster Medicine, Fukushima Medical University, Fukushima 960-1295, Japan; k-iyama@fmu.ac.jp; 2Department of Disaster Medicine, Fukushima Medical University Hospital, Fukushima 960-1295, Japan; yoshinob@fmu.ac.jp; 3Department of Hygiene and Preventive Medicine, Fukushima Medical University, Fukushima 960-1295, Japan; bamboo@fmu.ac.jp; 4Acute and Critical Care Center, Nagasaki University Hospital, Nagasaki 852-8501, Japan; kazy0722@nagasaki-u.ac.jp (K.Y.); tasaki-o@nagasaki-u.ac.jp (O.T.); 5Futaba Emergency Medical Support Center, Fukushima Medical University, Fukushima 960-1295, Japan; jshimada@fmu.ac.jp

**Keywords:** disaster, emergency responders, hazard, human resources, intent

## Abstract

To ensure human resource availability for a smooth response during various types of disasters, there is a need to improve the intent of those involved in responding to each hazard type. However, Disaster Medical Assistance Team personnel’s intent to engage with specific hazards has yet to be clarified. This study therefore aimed to clarify the factors affecting Disaster Medical Assistance Team members’ (*n* = 178) intent to engage with each type of hazard through an anonymous web questionnaire survey containing 20 questions. Our results show that the intent to engage in disaster response activities was significantly lower for chemical (50), biological (47), radiological/nuclear (58), and explosive (52) incidents compared with natural (82) and man-made hazards (82) (*p* < 0.01). Multiple regression analysis showed that incentives were the most common factor affecting responders’ intent to engage with all hazard types, followed by self-confidence. Thus, creating a system that provides generous incentives could effectively improve disaster responders’ intent to engage with specific hazards. Another approach could be education and training to increase disaster responders’ confidence. We believe that the successful implementation of these measures would improve the intent of responders to engage with hazards and promote the recruitment of sufficient human resources.

## 1. Introduction

The initial response by the Rapid Response Team or Medical Emergency Team is the most important factor associated with the prognosis of individuals in critical situations [1,2,3]. Over recent years, the general public has experienced various types of hazards, which can be divided into three groups, namely natural (e.g., earthquakes), man-made (e.g., vehicular accidents), and specific hazards (e.g., the coronavirus disease 2019 (COVID-19) or chemical terrorism), and the risk of facing these hazards, except traffic accidents, has not decreased over time [4,5,6,7]. Specific hazards can be further characterized as chemical (C), biological (B), radiological (R), nuclear (N), and explosive (E) (CBRNE) incidents, which clearly necessitate a rapid and smooth response to save people’s lives. The frequency of these hazards has been increasing, which has led to the need for proactive measures to be undertaken [8,9,10]. However, the Fukushima Daiichi nuclear power plant accident, one of the most famous radiological/nuclear (R/N) incidents, showed that providing smooth disaster response activities at all times can be difficult [11]. Therefore, advance planning is required when providing medical services in hazardous areas to facilitate the smooth implementation of disaster response activities.

The lack of human resources has been considered one of the major factors preventing smooth CBRNE incident response activities. Disaster responders have been known to have poor intent to engage in specific disaster response activities [12,13,14]. In fact, a number of surveys have revealed that even those who are willing to respond to natural hazards avoid being involved with nuclear or contagious disease incidents due to anxiety and lack of knowledge [13,14,15]. Our previous research also highlighted the poor intent to engage in CBRNE incidents [12]. This poor intent has been one of the major reasons for the lack of human resources, which has been the primary obstacle toward CBRNE incident response activities. Our previous study revealed that several factors, such as self-confidence, incentives, and family understanding, affect firefighters’ intent to engage in nuclear disaster response activities [16]. However, the intent to engage of disaster medical responders in CBRNE incident response activities has remained unclear.

The Great Hanshin–Awaji Earthquake of 1995 that occurred in Japan triggered the development of a disaster medical system and the establishment of the Disaster Medical Assistance Team (DMAT) to work during various disasters. The DMAT consists of physicians, nurses, and logistics staff as defined in the Basic Disaster Management Plan based on Japan’s Disaster Countermeasures Basic Act [17,18]. The DMAT currently plays a major role in large-scale disasters occurring throughout Japan. In other words, the DMAT is one of the most important disaster medical responders in Japan. Therefore, determining whether DMAT personnel would commit to engaging in acute-phase activities in the event of CBRNE incidents to smoothen the CBRNE incident response activities is certainly warranted.

Japan is the only country in the world with an experience of atomic bomb explosions. In addition, Japan experienced a large-scale nuclear accident in the Great East Japan Earthquake in 2011. In this study, we hypothesized that the intent to engage in CBRNE disaster response activities may differ between regions that have experienced nuclear accidents and regions that have not. The current study therefore conducted a questionnaire survey among two types of DMAT personnel: those who have experienced a nuclear disaster and those who have not. To ultimately smoothen each specific disaster response, the current study aimed to clarify the factors affecting the DMAT’s intent to engage in disaster response activities in order to determine future measures to improve engagement in disaster response activities.

## 2. Methods

An anonymous web questionnaire survey was conducted from October 2020 to November 2020. The questionnaire website URL was distributed to the mailing list of DMAT personnel assigned to a nuclear-disaster-affected area and those assigned to a nonaffected area. The response number was 204, with 178 effectively complete answers (effective response rate: 87.3%) finally being included in the analysis. The details of the study participants are presented in our previous study [12]. The questionnaire collected information on participants’ sex, age, occupation, family status, facility type, and experience in disaster response activities (Table 1). To validate the intent to engage in disaster response activities (natural, man-made, chemical, biological, radiological/nuclear, and explosive incidents), the following question was included: “Q1: Would you willingly actively engage in response activities during a *D* hazard? (where *D* is “natural,” “man-made”, “chemical”, “biological”, “radiological/nuclear”, or “explosive”)”. The participants were required to answer using the engagement intent score (EIS), which indicates their agreement to the abovementioned question (0–100%), with 100% indicating “strongly agree (yes)” and 0% indicating “strongly disagree (no)”. Respondents were also requested to indicate their level of agreement on a scale of 0–100% for the remaining 19 questions detailed in Table 2. The research model for investigating intent to engage was constructed based on the Theory of Planned Behavior [19,20]. The questions used in this study were based on content from previous studies [16,21].

The web questionnaire URL was distributed to two DMAT mailing lists; one is for a nuclear disaster-affected area and the other for a nonaffected area. A total of 204 personnel responded. After excluding 26 responses with missing data, 178 participants were ultimately included for analysis.

The analysis of age was conducted using a cutoff of 39 years, given that the mean age of DMAT personnel was 38.8 years old [12,22]. Scores for each hazard were compared using analysis of variance (ANOVA) and the Tukey–Kramer test for multiple comparisons. Multiple regression analysis, including background and responses to Q2–20, was conducted to determine factors affecting EIS (Q1). Factors with Variance Inflation Factor >4 were removed from the perspective of collinearity. We plotted factors with significant standardization coefficients (*p* < 0.05) into a path diagrammatic representation. All statistical analyses were performed using JMP 14 (SAS Institute Inc., Cary, NC, USA), with the significance level being set at 0.05.

## 3. Results

According to the primary outcome, the mean EIS (answers to Q1) for each hazard was as follows: natural (82.2), man-made (81.7), C (50.0), B (47.4), R/N (57.6), and E (52.4). After multiple comparisons, the proportions of natural and man-made hazards were significantly higher than C, B, R/N, and E incidents (all *p*-values < 0.01). Furthermore, R/N had a higher EIS than B (*p* < 0.05). For responses to Q2–Q20, our results show differences between the scores of six hazards, except for Q8, Q14, and Q20 (Appendix A). The results of multiple regression analysis are shown in Figure 1 and Figure 2 and Appendix B. Adjusted R^2^ for each model was as follows: natural (0.46), man-made (0.50), C (0.61), B (0.56), R/N (0.64), and E (0.67). As for the background factors affecting the intention to engage in a disaster, area, occupation, experience in natural disaster response activities, and experience in CBRNE incident response activities were the predominant influences (Figure 1). Being in a nonaffected area had a positive impact on the EIS for man-made hazards, whereas having experience with natural disaster response activities had a negative impact on the same. Moreover, having experience with CBRNE incident response activities had a negative impact on the EIS for C incidents, whereas being a nurse had a negative impact on the EIS for B incidents. Figure 2 shows the factors significantly affecting the EIS for each type of hazard in response to the Q2–20. Accordingly, Q6, Q7, Q16, and Q18 had a positive effect on the EIS for natural hazards. Q6 and Q12 had a positive effect on the EIS for man-made hazards. Q6 and Q7 had a positive effect on the EIS for C incidents. Q6 had a positive effect on the EIS in B incidents. Q2, Q3, Q6, and Q12 had a positive effect on the EIS for R/N incidents. Q3, Q5, Q6, Q12, and Q19 had a positive effect on the EIS for E incidents, whereas Q8 had a negative effect on the same.

## 4. Discussion

The current study was conducted for the ultimate purpose of smoothing out CBRNE incident response activities. To achieve this goal, we initially focused on the issue of securing sufficient human resources for better intervention, particularly focusing on the DMAT’s intent to engage in various type of disaster response activities. Although some studies have surveyed intent to engage in disaster response activities, most of them have been based on responses to questionnaires on a 2–10 scale. To the best of our knowledge, this has been the first survey to determine DMAT personnel’s intent to participate in specific hazards that scored intent on a continuous scale of 0–100%, which allows precise assessment of intent to engage in disaster response activities.

### 4.1. Factors Commonly Influencing Intent to Engage in Multiple Hazards

The three factors that had been previously identified to influence intent to engage, namely incentive, confidence, and family understanding, had a common and significant impact on multiple hazards in the current study [16]. Incentive (Q6) was found to have a common impact on all types of hazards, indicating that incentives are essential for effectively improving disaster responders’ intent to engage in various types of hazards. However, it is not the biggest influence on all hazards. For example, in man-made hazards, self-confidence (Q12) had the same degree of impact as incentive and, in explosive incidents, interest (Q5) had a higher impact than incentive. The next most common influencing factor was confidence (Q12), suggesting that being confident in the activity influenced responders’ intent to engage in man-made, R/N, and E incidents. These findings were similar to those presented in previous studies regarding firefighters’ intent to engage in R/N incidents [16]. Education and training during peaceful periods aimed at increasing the DMAT members’ confidence in engaging with hazards will also improve their intent to engage. On the other hand, although family understanding also significantly influence firefighters’ intent to engage in R/N incidents, the current findings show that family understanding (Q7) only influenced intent to engage in natural and C incidents. The subjective norm that one’s own occupation should engage (Q3) also had an effect on the intention to engage in R/N and E incidents, as these hazards are highly specific and occur relatively infrequently. While several previous studies on emergency medical service units have reported that being male and having years of experience influenced the readiness for the CBRNE response, the current study found no evidence of this [8].

### 4.2. Factors Affecting Intent to Engage That Are Unique to Each Hazard

#### 4.2.1. Natural Hazards

Our findings show that the need to educate and train oneself to satisfy the expectations of residents (Q16) and the need to actively participate in seminars (Q18) significantly influenced intent to engage in natural hazards. Among the six types of hazards mentioned herein, natural hazards were the most frequent. Therefore, the subjective norm of having to do something (Q16) and the attitude toward the behavior derived from that norm (Q18) were considered to be influential.

#### 4.2.2. Man-Made Hazards

The current study found that being in a nonaffected area and having experience in natural disaster response activities were positive and negative factors characteristically influencing intent to engage in man-made disaster activities. The difference between the nuclear-disaster-affected area and the nonaffected area, i.e., the survey sample for this study, not only reflects the difference in nuclear disasters but also, strictly speaking, other regional characteristics. For example, the number of traffic accidents per number of vehicles in the area is higher in the nonaffected area than in the nuclear-disaster-affected area [23,24]. Thus, the members of the nonaffected area may have been well-prepared for man-made hazards. In the case of natural hazards, DMATs are mainly dispatched to hospital facilities, but in the case of man-made hazards, DMATs are expected to be dispatched to the onsite field. Therefore, those who have been dispatched to respond to natural hazards may be negatively affected by the fact that they can easily imagine the horror of being dispatched to a man-made hazard site, even though there is a lot of confusion when being dispatched to a facility. On the other hand, for a CBRNE hazard, unlike a man-made hazard, it is not possible to even imagine the concrete scale and horror of the hazard, so, on the contrary, the experience of being dispatched for natural hazards does not seem to have an impact.

#### 4.2.3. Chemical Incidents

Our survey results show that having experienced CBRNE incident response activities was a characteristic factor negatively affecting the intention to engage in C incidents, suggesting that a single experience with CBRNE incidents may promote a subsequent decrease in intent to engage. In other words, the fact that they have experience with CBRNE means that they have knowledge of chemical incidents and can easily imagine how horrific they are, which may have a negative impact on the above results.

#### 4.2.4. Biological Incident

Being a nurse was herein identified as a negative factor affecting intent to engage in B incidents, which was thought to be attributed to the numerous roles and burdens nurses have carried in response to the recent COVID-19 pandemic. A survey by the Japanese Nursing Association revealed that not only the nurses who were involved in handling COVID-19, but also those who did not directly respond to COVID-19, were required to be reassigned to the department. Furthermore, one in five nurses responded that prejudice and discrimination against nurses due to COVID-19 exists [25]. Almost all nurses who responded felt stressed and anxious about COVID-19, which can negatively impact their mental health [26,27]. Our results suggest that nurses with a negative perception of the COVID-19 response may be less willing to engage in B incident response activities.

#### 4.2.5. Radiological/Nuclear Incidents

The desire to collect information regarding the hazard (Q2) and the belief that their own occupation should respond to the hazard (Q3) both had a positive effect on intent to engage in R/N incidents. Although R/N incidents have been widely recognized by the public due to the recent Fukushima accident, one can also assume that accurate information regarding the actual situation has not reached the public, which is necessary for DMAT members to receive [11]. Moreover, DMAT members are considered to have high subjective norms and a strong sense of mission for unknown hazards, such as R/N incidents. Since radiological and nuclear hazards are strictly distinct, different results may be obtained when they are separated. We expect further research to reveal the differences between these two hazards.

#### 4.2.6. Explosive Incidents

Factors positively influencing intent to engage in E incidents include the belief that their own occupation should respond to the hazard (Q3), interests (Q5), and the belief that exposure was acceptable as long as the responder’s family was safe (Q19). Similar to R/N incidents, subjective norm (Q3) had an impact on E incidents. It was also easy to understand why interest increased intent to engage. Although Q5 was excluded from the multiple regression in other hazards due to multicollinearity, the same trend can be inferred for other hazards. Among the CBRNE incidents, only E incidents had a significant effect on Q19, which may be attributed to the fact that only E incidents did not promote the spread of infection to or contamination of family members. While C, B, R, and N may cause damage to the surroundings due to secondary contamination, this is unlikely to be the case with E incidents [28,29,30]. However, belief in the indispensability of education and training (Q8) had a negative effect on intent to engage, which can be directly interpreted as that people with high intent to engage perceived education and training as unnecessary. However, given the cross-sectional nature of the current study, the causal relationship could unfortunately not be determined. In other words, this may indicate that people with low engagement intent perceive the need for education and training.

### 4.3. Limitations

Although the questionnaire website URL was sent to the registered e-mail addresses on the mailing list, we could not definitively determine whether all DMAT personnel had received it. Moreover, some people might have more than one e-mail address registered, whereas others may not have received them given that they had already changed their e-mail addresses. Therefore, the actual response rate remains unclear. This survey only involved DMAT personnel assigned to two areas. Given that not all DMAT personnel in other areas or occupations who might be engaged in various hazards were included in this study, further studies are warranted.

## 5. Conclusions

The current study reveals that DMAT members had lower intent to engage with CBRNE incidents compared to natural and man-made hazards. To smoothen the response to specific hazards in the future, measures to efficiently improve the currently low intent to engage with CBRNE incidents are required. Multiple regression analysis revealed that incentives are important to effectively improve DMAT members’ intent to engage with all CBRNE incidents. Therefore, establishing a system that provides generous incentives, including legal arrangements, will certainly be desirable in the future. Furthermore, given that self-confidence in the activity affected intent to engage with multiple hazards, training and education of DMAT members may be necessary to efficiently promote self-confidence in CBRNE incident response. Further research on specific measures should be expected in the future.

## Figures and Tables

**Figure 1 ijerph-18-11630-f001:**
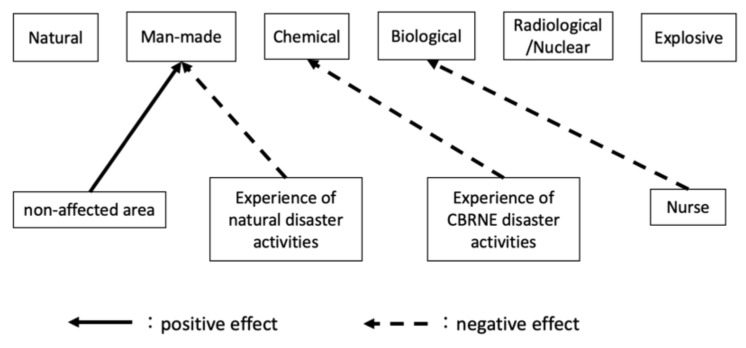
A path diagrammatic representation of the impact of background factors on intent to engage for each hazard. The solid and dashed lines represent a significant positive and negative impact on engagement intent score, respectively.

**Figure 2 ijerph-18-11630-f002:**
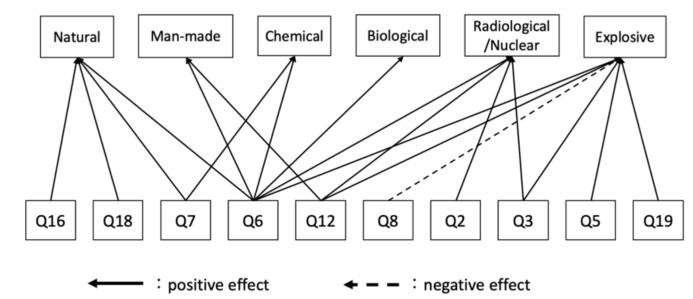
A path diagrammatic representation of the impact on intent to engage in each hazard based on survey questions. The solid and dashed lines represent a significant positive and negative impact on engagement intent score, respectively.

**Table 1 ijerph-18-11630-t001:** Participant characteristics.

	*n* = 178
**Area, *n* (%)**	
Nuclear-disaster-affected area	79 (44.4)
Nonaffected area	99 (55.6)
**Sex, *n* (%)**	
Female	128 (71.9)
Male	80 (28.1)
**Age (years), *n* (%)**	
20−29	14 (7.9)
30−39	70 (39.3)
40−49	67 (37.6)
Over 50	27 (15.2)
**Occupation, *n* (%)**	
Physician	41 (23.0)
Nurse	69 (38.8)
Other medical staff	26 (14.6)
Administrative staff (nonmedical)	42 (23.6)
**Family, *n* (%)**	
With	135 (75.8)
Without	43 (24.2)
**Disaster base hospital, *n* (%)**	
Yes	141 (79.2)
No	37 (20.8)
**Experience in natural disaster response activities, *n* (%)**	
Yes	123 (69.1)
No	55 (30.9)
**Experience in CBRNE disaster response activities, *n* (%)**	
Yes	14 (7.9)
No	164 (92.1)

*n*, number; CBRNE, chemical, biological, radiological, nuclear, and explosive.

**Table 2 ijerph-18-11630-t002:** Questionnaire contents.

Q1	Would you Willingly Actively Engage in Response Activities during a *D* hazard?
Q2	Would you willingly collect information preparing for a *D* hazard?
Q3	Do you think that your occupation should actively response to a *D* hazard?
Q4	How much opportunities do you have to learn about *D* hazards in your environment?
Q5	How much interest do you have in *D* hazards?
Q6	Will you engage in *D* disaster response activities if there are incentives, such as insurance and special salaries?
Q7	How much do you think your family will understand about your activity during a *D* hazard?
Q8	Do you think that education and training are indispensable for *D* disaster response activities?
Q9	If your colleagues are preparing for *D* hazards (e.g., education or training), do you think you should take action as well?
Q10	Does your workplace offer seminars on *D* disaster response?
Q11	How often do you think a *D* hazard will occur in your area?
Q12	Do you have self-confidence in *D* disaster response activities?
Q13	Are you anxious about the activities in a *D* hazard situation?
Q14	Do you feel sorry to your family if you are exposed to *D* hazards?
Q15	Would you willingly actively work on *D* hazard countermeasures?
Q16	Do you think that your own occupation should be routinely educated and trained on *D* hazards to meet the expectations of citizens?
Q17	At your own workplace, is it easy to obtain information about seminars for *D* disaster response?
Q18	Would you willingly actively participate in seminars on *D* hazards?
Q19	If your family is safe, can you be exposed to *D* hazards during a disaster response?
Q20	How much do you think it is important to prepare for a *D* hazard (e.g., education or training)?

*D* can be replaced with “natural”, “man-made”, “chemical”, “biological”, “radiological/nuclear”, or “explosive”.

## Data Availability

No additional data are available for this study. However, inquiries concerning the data may be addressed to the corresponding author.

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
