# Peer review of "Increasing Disaster Medical Assistance Teams’ Intent to Engage with Specific Hazards"

_ijerph, 2021, doi:10.3390/ijerph182111630_

Round 1
Reviewer 1 Report
The manuscript is well written, and it appears to be the result of a comprehensive work
- Originality/Novelty: Question is well defined as an evaluation tool for nurses in disaster situations.
- Scientific Soundness:sampling in various specialties may bring additional information to the findings.
- Interest to the Readers: I believe the information in this abstract is of interest to all nurses and health care professionals and will be a great addition to nursing literature.
- Overall Merit: I believe publishing this abstract will be beneficial to all health care professionals. Possibly utilizing this information in other countries and revising the tool for use in various situations will bring benefit and increase readership in this area.
Some minor comments
I think some tables 2-4 could be added as a Appendix. You do not need to add it in the main text. Or consider changing 3-4 into figures.
Term “natural disasters” + 4.2.1 the UNDRR agreed last year that there is no such thing as a natural disaster so we dissuade people from using the term (there are natural hazards, and there are disasters).
4.2.5 Please split this section into to independent. There is serious difference between R and N
4.2.6 This is a little but confused. We can talk about explosive materials but term explosive disasters is not really common
Another thing that should be enchanted is the lack of references. Number of 15 is not enough to support the thesis especially in discussion.
I would like to see some more. If you referring to CBRNE please cite
http://dx.doi.org/10.5281/zenodo.5553076
Reviewer 2 Report
Line 39-40 and 42-43 Is the risk of facing all the three categories of disasters increasing? Provide evidence. Climate related natural disasters (floods, cyclones etc) seem to be increasing with climate change, but geological disasters (e.g. earthquakes)? And man-made disasters (e.g. vehicle accidents)? Are the risks of these increasing?
Line 50-51 What concrete evidence is there that disaster responders have been known to have poor intent to engage in specific disaster response activities?
Line 71-72 This study would be vastly improved if the survey was not limited to simply two groups (those who have experienced a nuclear disaster and those who have not).
Line 94 Figure 1 is absolutely not necessary.
Line 86-87 (and all the questions in the survey) I think the original survey would have been conducted in Japanese, and Table 2 is a translation into English by the authors of the questions in the survey. The exact wording of the questions and their meaning/understanding is absolutely critical here. For example, were the words “Do you like to actively engage in response activities during a D disaster” the actual intent? I am sure the intent was more like “Would you willingly actively engage in response activities during a D disaster”. But maybe not. And the response will reflect that. Please be very clear about this, because the words mean everything here! For this reason this paper might be better suited to a Japanese-language journal.
Lines 113-114 Carrying on from my previous point. The responses will depend on the exact wording of the question. This is likely to impact (even invalidate ALL of the results).
Line 98 Table 2 This survey examines a subset of potential reasons that DMAT personnel may or may not engage. Please explain why this set of factors is exhaustive and complete.
Line 146 What are the R2 values and significance levels of the multiple regressions. They are a key part in explaining the efficacy of the model.
Lines 156-159 I cannot see why a continuous scale is any better than a 1-10 scale for the purposes of DMAT personnel’s intent to participate in specific disasters. Please explain.
Line 163-164 Humans are net value maximisers. They would always want more value rather than less. The way Q6 is phrased it is obvious that DMAT personnel would respond positively for all types of disasters.
Line 196-208 “There may also be regional differences in the backgrounds of DMAT members…..” This whole section is conjecture. Provide evidence or delete.
Line 217-221 Provide evidence or delete.
Line 226-229 Provide evidence or delete.
Line 237-242 Provide evidence or delete.
Round 2
Reviewer 2 Report
I would suggest that the article be reviewed by a native English speaker to make it even better
Author Response
Thank you for pointing out the concern about language. Our manuscript has been proofread for the English language by Enago (www.enago.jp). We have changed the acknowledge and funding information according to the above-mentioned fact.
